# Validity and Reliability of Movesense HR+ ECG Measurements for High-Intensity Running and Cycling

**DOI:** 10.3390/s24175713

**Published:** 2024-09-02

**Authors:** Raúl Martín Gómez, Enzo Allevard, Haye Kamstra, James Cotter, Peter Lamb

**Affiliations:** 1School of Physical Education, Sport and Exercise Sciences, University of Otago, Dunedin 9054, New Zealand; 2Auckland Bioengineering Institute, University of Auckland, Auckland 1010, New Zealand; 3Department of Human Movement Sciences, Vrije Universiteit Amsterdam, 2025 Amsterdam, The Netherlands

**Keywords:** Movesense HR+, Garmin HRM-Pro, ECG measurements, wearable devices, R-peak detection, sports science, validity, reliability, exercise intensity, heart rate variability (HRV)

## Abstract

Low-cost, portable devices capable of accurate physiological measurements are attractive tools for coaches, athletes, and practitioners. The purpose of this study was primarily to establish the validity and reliability of Movesense HR+ ECG measurements compared to the criterion three-lead ECG, and secondarily, to test the industry leader Garmin HRM. Twenty-one healthy adults participated in running and cycling incremental test protocols to exhaustion, both with rest before and after. Movesense HR+ demonstrated consistent and accurate R-peak detection, with an overall sensitivity of 99.7% and precision of 99.6% compared to the criterion; Garmin HRM sensitivity and precision were 84.7% and 87.7%, respectively. Bland–Altman analysis compared to the criterion indicated mean differences (SD) in RR’ intervals of 0.23 (22.3) ms for Movesense HR+ at rest and 0.38 (18.7) ms during the incremental test. The mean difference for Garmin HRM-Pro at rest was −8.5 (111.5) ms and 27.7 (128.7) ms for the incremental test. The incremental test correlation was very strong (*r* = 0.98) between Movesense HR+ and criterion, and moderate (*r* = 0.66) for Garmin HRM-Pro. This study developed a robust peak detection algorithm and data collection protocol for Movesense HR+ and established its validity and reliability for ECG measurement.

## 1. Introduction

Innovative technological applications have been a focal point in sports science research, with implications extending to elite sports analysis, injury surveillance, and clinical evaluations. Among these technological advancements, wearable devices have emerged as promising tools due to their portability, affordability, and ease of use. Movesense, a Finnish product platform initially developed by Suunto, offers a versatile sensor solution with an open Application Programming Interface (API) and Software Development Kit (SDK), allowing developers to create custom wearable devices and applications. The Movesense HR+ device contains a variety of sensors including an accelerometer, gyroscope, magnetometer, heart rate monitor, and temperature sensor.

Despite its potential, the validity of Movesense HR+ electrocardiogram (ECG) data relative to traditional ECG systems remains unreported. Prior research has demonstrated the correlation of the Movesense Medical sensor with a 12-lead ECG for heart rate and its variability during rest and indoor stationary cycling conditions [1,2]. Additionally, studies using the Movesense Medical device for atrial fibrillation (AF) detection have shown promising results. One study demonstrated that the Movesense chest strap ECG could effectively detect AF with sensitivity and specificity comparable to three-lead Holter ECGs [2]. Another study highlighted the feasibility and accuracy of a consumer-grade Movesense heart belt for AF detection, showing high sensitivity, specificity, and user preference over traditional Holter monitors [3]. The Movesense medical sensor is registered as a Class IIa medical device accessory. The Movesense HR+ offers a more financially economical alternative (Movesense HR+ priced at EUR 119 compared to the medical device at EUR 349 [4]). Although it has been successfully used for arrhythmia detection [5], and the accuracy of the automatically calculated HRV data from the device have been compared against a sport vest [6], no reliability or validity analysis has been reported on its ECG tracking, nor have either of the Movesense models been assessed for a more physically demanding activity such as running—specially, a high-intensity running protocol where data collection with wired systems can be hampered by cables shaking.

The primary aim of this study was therefore to compare the validity and reliability of Movesense HR+ ECG measurements across different exercise modes and intensities, using criterion measurements from a standard three-lead ECG system. We seek to establish whether Movesense HR+ can provide accurate and consistent data in both resting and physically demanding conditions, contributing to its potential utility in sports and clinical settings. A custom software environment (Movesense firmware, Android apps and post-hoc signal processing software) was used to perform all required analyses and ensure full transparency regarding the software’s processing of raw data. The present study therefore examined the reliability and validity of the Movesense HR+ ECG measurements incorporating an R-peak detection algorithm and an ECG data collection routine. The bespoke system is built on customised Movesense firmware, with a freely available Android app developed in-house. A secondary aim of this study was to perform a reference comparison for the industry-leading device, Garmin HRM-Pro, against the ECG criterion measurements.

## 2. Materials and Methods

### 2.1. Participants

Twenty-one healthy adults were the participants; fourteen undertook a running incremental protocol (male: *n* = 9, female: *n* = 5; age: 22 ± 3 years), and seven undertook a cycling incremental protocol (male: *n* = 5, female: *n* = 2, mean age: 24 ± 3 years). All participants reported an absence of pre-existing conditions affecting the autonomic nervous system, cardiovascular system, and the lower limbs within the past three months. Informed consent was obtained from all participants in writing, and the study protocols were approved by the University of Otago Human Ethics Committee (approval number H23/020).

### 2.2. Experimental Protocol

Before each session, participants provided written, informed consent. Participants’ body mass and height were then measured. Subsequently, they underwent an incremental ramp protocol until voluntary exhaustion on a treadmill or a cycle ergometer. Before and after the incremental test, participants had a 5-min rest period. Consequently, the experimental protocol consisted of three activities, as detailed in Table 1. During the incremental phase, subjects reported their Rating of Perceived Exertion (RPE) using the linear Borg scale, helping to ensure that exhaustion was reached. The choice of a step-incremental test over a rectangular test was made to assess HR and heart rate variability (HRV) evolution across various exercise intensities and modes, with a focus on characterizing Movesense HR+ device performance.

The placement of chest straps (Movesense HR+ and Garmin HRM-Pro) was not randomised in this study. Movesense HR+ was consistently placed under the nipples for all participants, as shown in Figure 1. However, the placement of Garmin HRM-Pro varied depending on participant comfort and the feasibility of positioning it above or below Movesense HR+, especially with female participants wearing sports bras. In preparation for the experiment, skin was cleaned with alcohol before attaching the 3-lead electrodes. No glue was used for Movesense or Garmin chest straps; instead, the electrodes were wetted with water to recreate the most common use of the devices. The study also involved participants wearing respiratory face masks during the exercise sessions; however, respiratory data were not used for the current study.

### 2.3. Materials

The study utilized a treadmill (model: STEX 8020T) and a cycle ergometer (model: Lode Excalibur Sport) for exercise. Materials and methods used for data collection are shown in Figure 1. ECG data collection involved two systems: a 3-lead ECG setup (ADInstruments, Dunedin, New Zealand) and the Movesense HR+ single-channel ECG with a chest belt (Movesense, Vantaa, Finland). The 3-lead ECG system recorded data at 1000 Hz using hardware (Powerlab, ADInstruments) and software (Labchart 8.1.24, ADInstruments), while Movesense HR+ was sampled at 500 Hz. A Garmin chest device (Garmin HRM-Pro) connected to a Garmin Fenix 3 watch was used to compare metrics related to HR and HRV (Table 2). Both the Movesense HR+ and Garmin HRM-Pro chest straps use two electrodes to measure the electrical activity of the heart, which is then transmitted to the respective devices for processing and analysis. 

### 2.4. Data Collection

The 3-lead ECG data were saved in a .txt file format, with each row representing an ECG voltage value accompanied by timestamps at the beginning of the recording. ECG data from the Movesense HR+ sensor were captured using purpose-built and freely available FreeLab [7] and GetLabData [8] applications for Android; see Figure 2. Before data collection, all Movesense HR+ devices used in the research were updated with customised firmware, which enabled specific services and characteristics for transmitting ECG and IMU data from the sensor via a Bluetooth Low Energy (BLE) connection [9]. The customised version primarily utilised Primbs’ [10] features, adapting the default values and characteristics to the specialised needs of our research. This bespoke adaptation involves changing the default values to the maximum frequencies, 500 Hz for ECG (and 208 Hz for IMU, although these data are not included in the current study) and increasing the disconnection time to 120 s. The recorded data were sent by the device in packages of 16 ECG values (in µV), along with a relative timestamp indicating the time elapsed since the activation of the device, and stored in a .csv file format upon reception. Similar to the 3-lead ECG data, a global timestamp was included at the start of the recording for synchronisation purposes, both global timestamps were converted to epoch, Unix time [11]. Garmin data were downloaded in .fit file format from Garmin Connect and subsequently converted to .csv files [12]. Each .csv file contained HR and HRV values, with both relative and global timestamps added during processing. We note that Garmin’s global timestamp is not a conventional Unix epoch timestamp, representing the number of seconds elapsed since 1 January 1970 (midnight UTC/GMT) [11]. Instead, it utilises a “fit timestamp”, defined by the FIT Profile as an uint32, representing the number of seconds since midnight on 31 December 1989, UTC [13]. This date is often referred to as the FIT Epoch. Consequently, a conversion of the Garmin global timestamp was required by adding 63,106,560,000 ms.

The ECG data from the 3-lead and Movesense HR+ device were consequently processed [5] using the self-created peak detection algorithm; see Figure 2. The algorithm for R-peak detection involves several steps to ensure accurate and reliable peak identification during exercise. The first step is to divide the ECG data into intervals, calculating the data frequency and managing the intervals based on this frequency. Next, two independent peak detection processes are performed for each interval: one for R peaks and one for S peaks, with the ECG data inverted for the latter. Each detection method involves sorting the data by voltage and iteratively determining a threshold for peak detection by analysing the highest voltage values. After identifying the R and S peaks, the results from both methods are combined to form a comprehensive list, filtering and merging peaks detected by both methods. Subsequently, the algorithm performs a statistical analysis of the detected peaks identifying abnormal RR intervals. For these intervals, new potential peaks are detected by analysing subsets of ECG data around target timestamps and calculating associated characteristics. The algorithm iteratively optimises its parameters, recalculates heart rate and heart rate variability, and refines the detection process to maintain high accuracy. Finally, the detected peaks are compiled, and additional statistical analyses are conducted to ensure data quality. This comprehensive approach allows the algorithm to handle the complexities of ECG signal processing during physical activity effectively, focusing on providing robust and accurate results for further analysis.

A free executable version of the algorithm is available through a GUI in the resource folder [9]. Peak detection was verified through visual inspection of the ECG time-series plots and identified peaks, ensuring the accuracy of R-peak detection.

The global timestamps from the three data sources were synchronised using a custom algorithm; see Figure 2. This algorithm used the global timestamps of each heartbeat and the resulting RR’ values to align the data streams. Data were also visually inspected to verify the correct alignment across different devices. For each participant and activity, a final set of three synchronised files was obtained; see Figure 2.

No resampling or filtering was applied to the collected data. Therefore, the R-peak detection, data analysis and subsequent statistical analysis were conducted using the original sampling frequencies: 1000 Hz for the 3-lead ECG and 500 Hz for the Movesense HR+ ECG (not relevant for Garmin as only RR’ intervals can be exported [14]).

### 2.5. Statistical Analysis

The statistical analysis was comprised of two steps. Firstly, it involved assessing the sensitivity and precision of the peaks detected by the tested devices in comparison to the peaks from the criterion, 3-lead ECG; see Figure 2. Secondly, the analysis involved assessing the agreement of the RR’ values derived from the detected peaks.

The sensitivity and precision formulas are calculated as follows [15,16] (Figure 2):

True Positive (*TP*): The model correctly identifies the positive class. 

False Negative (*FN*): The model incorrectly identifies the negative class for a positive instance, also known as a Type II Error. 

False Positive (*FP*): The model incorrectly identifies the positive class for a negative instance, also known as a Type I Error.
$$Sensitivity=\frac{TP}{TP+FN}\cdot 100$$
$$Precision=\frac{TP}{TP+FP}\cdot 100$$

To classify a peak detected with the correspondent device (Movesense HR+ or Garmin HRM-Pro) as a *TP*, the R-peak timestamps from the criterion measurement and the corresponding device timestamps were compared. The number of R-peaks found by the device within a 100 ms threshold were counted and considered to have found the same R-peak. While other authors suggest a 75 ms window [17], a wider threshold of 100 ms was adopted considering the 200 ms refractory period between QRS complexes [18] and the fact that the frequency of the sensor from the Garmin device is unknown and Garmin R-peaks were calculated based on the RR’ intervals from the .fit file [14].

Following the RR’ interval study, two additional statistical analyses were conducted. One ascertained the correlation of the heart rate derived from the RR’ values across various intensities of the incremental test and calculated the intraclass correlation coefficient type single fixed rates (ICC3) to assess the reliability of the average heart rate (HR) across all exercise intensities for each device compared to the 3-lead ECG. The other analysis was of the devices’ capability to measure HRV at rest, by comparing the resulting standard deviation (SD) of their RR’ interval with that obtained from the 3-lead ECG.

## 3. Results

All data, including the corresponding R-peaks and the derived RR’ intervals, were visually inspected to ensure accuracy and remove artifacts. Only data accurately received, with good connection across devices, for each participant and activity were included in the statistical analysis. For Movesense, data not used due to poor connection were indicated by almost absent voltage values and a flat signal. For the three-lead system, disconnection of one or more leads results in a total absence of ECG peaks, detectable by the peak detection algorithm based on resulting abnormal peaks. Disconnection in Garmin HRM-Pro leads to HRV RR’ interval values of 5000, indicating a non-detected peak in the previous 5 s. When a disconnection or signal error occurred for any participant or activity, the corresponding file was cut at that point, and only the data before the disconnection were used.

Twenty-one participants completed Activity 1 (Rest). One participant could not proceed to Activities 2 and 3 because of connection problems with the BLE devices. Among the twenty participants who completed all three activities, data from two runners could not be used for the incremental test analysis: one due to problems with the three-lead sensor connection resulting in excessive signal noise, and the other because of noise from the Movesense and Garmin signals indicating poor sensor connection. For Activity 3, data from one participant (also a runner) could not be used due to noise in the three-lead, Movesense, and Garmin devices. Finally, in the analysis of Activity 1 with Garmin, data from two participants were excluded due to noise throughout the resting period.

Runners showed lower average heart rates at rest during Activity 1 compared to cyclists. However, during the incremental test (Activity 2) and final rest period (Activity 3), runners achieved higher mean and maximum heart rates compared to cyclists (Table 3).

### 3.1. Sensitivity and Precision of the Peaks Detected

The sensitivity and precision of peak detection systems were evaluated across various activities. Across all activities and participants, a total of 81,858 heartbeats were analysed. Movesense HR+ demonstrated an overall sensitivity of 99.66% and precision of 99.58%, compared to the criterion three-lead ECG. Table 4 presents the detailed results; the Movesense HR+ sensitivity and precision were greater than 99% for all activities and modes of exercise. With a total TN of 38,059,993 for the three activities, the resulting overall DER, specificity and accuracy of Movesense HR+ were 0.002%, 99.999% and 99.998%, respectively.

Similarly, we determined the sensitivity and precision for Garmin HRM-Pro. Combining both resting activities, Garmin HRM-Pro exhibited a sensitivity and precision of 95.83% and 95.88%, respectively. The overall analysis of 76,590 heartbeats across all activities yielded a sensitivity of 84.81% and precision of 87.79% (Table 5). With a total TN of 38,059,993 for the three activities, the resulting overall DER, specificity and accuracy are 0.002%, 99.999% and 99.998%, respectively (Figure 2). With a total TN of 37,454,530 for the three activities, the resulting overall DER, specificity and accuracy of Garmin HRM-Pro, were 0.055%, 99.969% and 99.945%, respectively.

### 3.2. Agreement of the RR’ Values Derived from the Detected Peaks

The agreement between corresponding RR’ intervals obtained through Movesense HR+ and Garmin devices compared to the criterion system was assessed. Given that RR’ intervals serve as the foundational metric for subsequent HRV calculations, examining the mean difference and standard deviation (SD) between these intervals is important in determining the suitability of these devices for generating accurate and valuable derived metrics. To this end, Bland–Altman plots are shown covering the different activities; see Figure 3.

Considering that the mean values of the RR’ intervals of each activity are 887.8 ms, 400.5 ms, and 574.3 ms for Activities 1, 2 and 3, respectively, the mean differences of 0.23 ms, 0.38 ms, and 0.47 ms between Movesense HR+ and the three-lead ECG obtained for each activity represent a percentage of 0.03%, 0.09% and 0.08% of the RR’ mean values. For Garmin, the mean differences of 8.48 ms, 27.69 ms, and 13.93 ms represent a percentage of −0.96%, 6.91% and 2.43% for Activities 1, 2 and 3.

### 3.3. Correlation of the Heart Rate Derived from the RR’ Values across Exercise Intensity

Figure 4 shows the resulting plots of the HR correlation analysis across all participants and types of exercise; the figure includes all heartbeats throughout the incremental test for the (a) Movesense HR+ and (b) Garmin HRM-Pro devices. The observed lower sensitivity of the Garmin HRM-Pro, particularly at high intensity (Table 5), is evident in a vertical block of points between 180 and 200 beats per minute (bpm) of the three-lead ECG. This block corresponds to Garmin values ranging from lower than 50 to 180 bpm, primarily due to missing peaks and incorrect RR’ interval values. The diagonal lines with a lower incline in Figure 4b represent instances where Garmin missed peaks with a frequency, for example, missing one out of every two peaks, resulting in a heart rate measuring half the real value of the three-lead. As demonstrated in one of the lines, a heart rate of 50 bpm from Garmin equates to 100 bpm for the three-lead ECG, while a heart rate of 100 from Garmin corresponds to approximately 200 bpm for the three-lead ECG.

To quantify this agreement, ICC3, ICC-type single fixed rates were calculated to assess the reliability of the average HR across all exercise intensities compared to the three-lead ECG. The ICC3 for the Movesense HR+ was 99.98%, indicating near-perfect agreement with the criterion. The Garmin HRM achieved an ICC3 of 95.95%, highlighting a good level of agreement, but with slightly more deviation compared to the Movesense HR+.

### 3.4. Comparative Analysis of RR’ Interval Results: Graph Samples

Figure 5a shows an example of Movesense peak detection aligned with the three-lead RR’ results. Similarly, Garmin Figure 5c also demonstrates good agreement in the sample, albeit with the exception of some incorrectly detected peaks, resulting in shorter RR’ intervals in case of an incorrectly added peak, or longer intervals when a peak is missing. In Figure 4, the alignment of the correspondent device signal with the three-lead signal from the synchronization algorithm can be observed.

Figure 5b,d from Activity 2 demonstrate excellent agreement between the RR’ intervals obtained from Movesense HR+ ECG and the three-lead ECG, with only 4 peaks missing out of a total of 3270 observed during the incremental test. In particular, the accuracy exhibited during the most intense phase of the exercise in the incremental test is very high. In contrast, Garmin, while initially matching the criterion RR’ results at the beginning of the test, shows errors in RR’ intervals during the higher-intensity phases. This is evidenced by missed peaks leading to higher RR’ values. Notably, during the cool-down period following the most intense phase of the incremental test, the Garmin device demonstrates a recovery in its capacity to detect peaks with higher accuracy.

### 3.5. Capacity of the Devices to Measure the HRV at Rest by Comparing the Resulting Standard Deviation (SD) of Their RR’ Interval with the One Obtained with the Three-Lead ECG

This section investigates the capacity of the devices to measure HRV at rest by comparing their SD of RR’ intervals with that obtained from the three-lead system. Figure 6 shows correlation and Bland–Altman plots of the Movesense HR+ and Garmin HRM-Pro against the three-lead system for all participants. Considering that the mean value of the SD of the RR’ for activity 1 was 95.54 ms, the mean difference of −0.92 ms between Movesense HR+ and the three-lead system represent a percentage of −0.96%. For Garmin, the mean difference of –22.7 ms represents a percentage of −23.76%.

## 4. Discussion

### 4.1. Accuracy and Reliability of Movesense HR+ ECG Data

The aims of this study were to establish the accuracy and reliability of Movesense HR+ ECG data across various exercise modes and intensities. Our results indicate the promising potential of Movesense HR+ ECG as a viable alternative for accurate ECG data recording during both rest and exercise. The consistently high sensitivity and precision values obtained across all participants and activities underscore the Movesense HR+’s capacity to serve as a cost-effective and user-friendly tool for both research and individual use. The high level of accuracy demonstrates the effectiveness of this portable BLE device in collecting high-quality data even during intense physical activity. The portability feature and relatively low cost are particularly important for various applications, including field research, coaching, and clinical settings, where easy-to-use, portable devices that can maintain data accuracy are essential. Moreover, the use of portable BLE devices can help transition research from controlled lab environments to real-world settings, improving the reliability and applicability of metrics obtained.

Agreement between Movesense HR+ and three-lead ECG was excellent and superior to that of the Garmin HRM-Pro. Research on the validity of devices such as the Movesense Medical device rely on comparing SD and other derived HRV values against the criterion rather than directly assessing peak detection sensitivity and precision. Movesense HR+ exhibited a strong correlation with three-lead ECG for RR’ SD during activity 1, initial rest, which is consistent with previous validation studies on the Movesense Medical device [1].

The capability of devices to accurately measure instantaneous heart rate from RR’ intervals under differing levels of physical exertion is important for coaches, athletes and practitioners. The incremental test, particularly during moments of higher intensity, poses challenges for heart rate monitoring devices due to increased movement intensity and the potential for sensor disconnections. Additionally, the higher frequency of peaks per second during intense periods, reaching three to four times the number observed at rest, further challenges the devices’ tracking abilities. Compared to Garmin HRM-Pro, the agreement between Movesense HR+ ECG and three-lead ECG based on RR’ intervals, particularly during intense exercise phases, was excellent. Garmin HRM-Pro encountered difficulties in detecting peaks accurately during high-intensity phases, often leading to higher RR’ values. The considerable difference between the Pearson correlation and ICC3 results for the Garmin HRM-Pro suggests that individual effects may impact the overall results, indicating some variability in individual measurements, while Movesense HR+ exhibited consistently high r and ICC3 values.

One of the benefits of the FreeLab Android app when collecting ECG data from the Movesense HR+ devices is the capacity to plot the ECG in real time, together with an R-peak algorithm for real-time detection. This functionality allows for checking of the waveform of the received ECG data during data collection. This real-time waveform checking not only ensures accuracy after collecting the information but also enables monitoring during data collection to verify that the sensor is correctly placed, receiving the signal, and that the BLE connection is working well. In contrast, with Garmin, it is only possible to perform real-time monitoring of the correct reception of the classic, average heart rate; the RR’ intervals are accessible only after analysing the resulting .fit file. Real-time peak detection by chest strap devices, transmitting information obtained with their sensors and algorithms to a storing data device, such as a watch, limits the possibility of improving the calculated RR’ intervals. While software tools used to analyse RR’ data from .fit files offer correction options [19], including percentage-based or device-specific corrections, the ability to manually verify R-peak detection and derived RR’ values remains a distinct advantage of ECG voltage-based systems like Movesense HR+ ECG. This flexibility enables the refinement of algorithms and even the manual selection of peaks over the ECG waveform, enhancing the accuracy of results. The implementation of the FreeLab software environment, customised firmware, Android apps, and Python GUI has demonstrated its potential to support peak calculations and further analysis. The ability to synchronize ECG data collected from Movesense or a three-lead system with HRV/RR’ data from .fit files (e.g., Garmin’s .fit files) enhances its utility for research analysis and comparison studies [2,3,5,6]. 

During intense exercise, challenges related to the accuracy of wearable devices often arise [20,21]. Factors such as motion artifacts, changes in skin contact, and signal interference can impact the reliability of heart rate monitoring and RR’ value capture. For example, increased movement during high-intensity activities can lead to inaccuracies in heart rate measurements, particularly with wrist-based sensors. Additionally, sweat and moisture build-up can affect the electrode–skin interface, potentially resulting in erratic readings. To mitigate these challenges and ensure accurate data collection, it is recommended to use chest straps or armbands, which offer better stability and contact with the skin, especially during precise heart rate training sessions. Studies have shown that chest straps and armbands provide more consistent and reliable heart rate monitoring, particularly during intense physical exertion [22]. This study demonstrates how the combination of new devices with appropriate software can overcome these difficulties, opening new possibilities for the use of wearable devices to collect accurate and reliable metrics.

### 4.2. Limitations

The sample size was small, consisting of young, healthy participants, which may limit the generalisability of our findings. Additionally, due to the inability to place both chest straps (Movesense HR+ and Garmin HRM-Pro) in identical positions, the results may be subject to bias based on the location of the chest strap. Furthermore, some data were excluded due to connectivity issues, particularly during or after activities involving intense movement. It is important to note that the placement of chest straps was not randomised in this study; Movesense HR+ was consistently placed under the nipples, as shown in Figure 1, while Garmin HRM-Pro was positioned above or below Movesense HR+ depending on participant comfort, due to limitations with sports bras, for example. Despite selecting only sets of data from Garmin where the connection was correct, this limitation may impact the accuracy of results obtained from the Garmin HRM-Pro. Additionally, the study focused primarily on running and cycling, and data from other modes of exercise were not tested, which could further limit the generalisability of our findings to a wider range of physical activities.

## 5. Conclusions

Our evaluation of Movesense HR+ ECG data, coupled with the flexibility and potential for refinement offered by ECG voltage-based systems, highlights its potential as a portable reliable tool for accurate ECG data recording across various exercise modalities. These findings have significant implications for both research studies and individual health monitoring as well as the development of enhanced physiological assessments of human performance. Users can make use of the knowledge gained from this study to further develop their use of Movesense HR+ and other ECG-based wearable devices. This includes ensuring optimal sensor placement, utilising real-time monitoring features to verify data accuracy during collection, and employing post-processing software for accurate calculation and validation of metrics derived from the raw ECG data, especially during intense exercise. Furthermore, from this study, we can underline the possible limitations of commercial devices, particularly during intense exercise, in accurately detecting the R-peaks and calculating the RR’ intervals in real time.

## Figures and Tables

**Figure 1 sensors-24-05713-f001:**
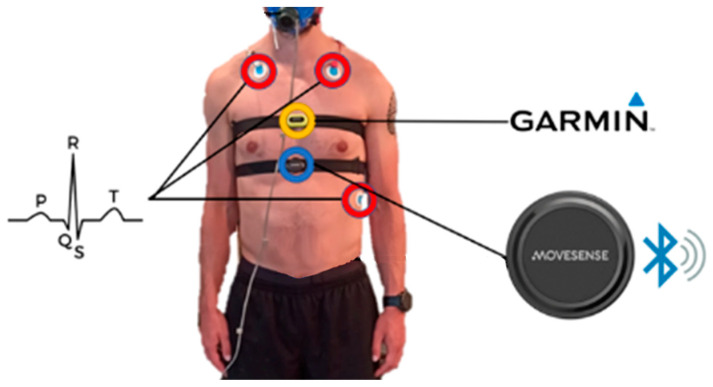
Configuration of 3-lead ECG, Movesense and Garmin devices. Participants were also equipped with a respiratory face mask; however, respiratory data were not included in the current study.

**Figure 2 sensors-24-05713-f002:**
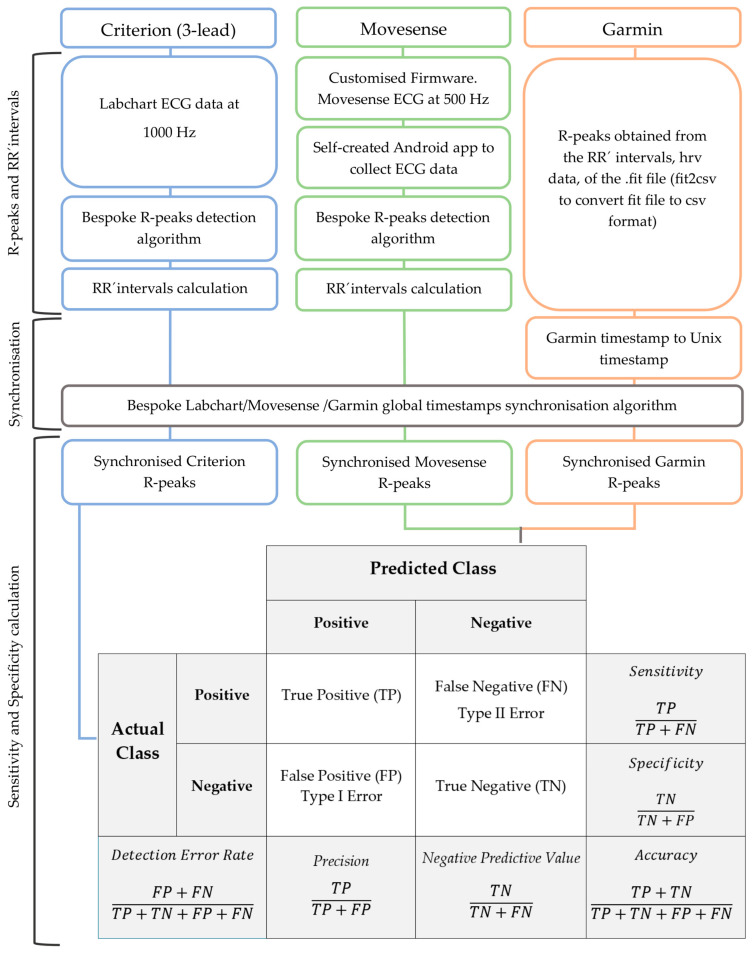
Diagram illustrating the data collection and statistical analysis process.

**Figure 3 sensors-24-05713-f003:**
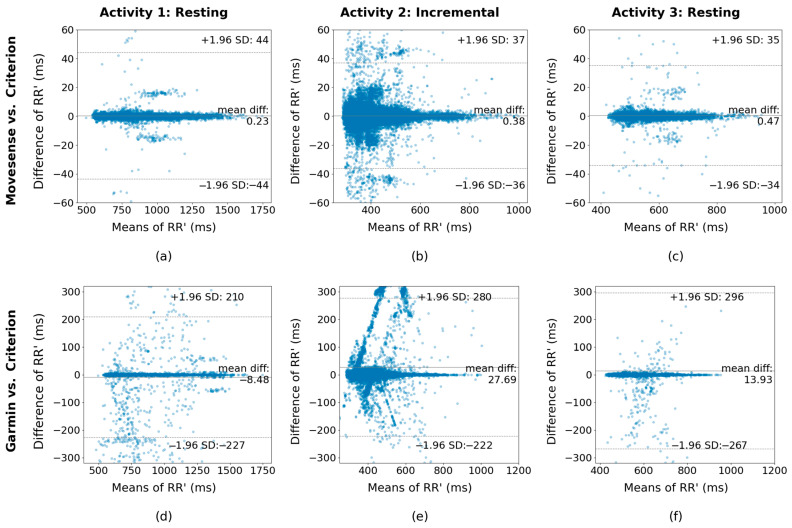
Bland-Altman plots of RR’ intervals derived from the detected peaks using Movesense HR+ and criterion for (**a**) Activity 1, initial 5 min rest; (**b**) Activity 2, incremental test; (**c**) Activity 3, final 5 min rest; and Bland-Altman plots of Garmin HRM-Pro RR’ vs. 3-lead RR’ intervals for (**d**) Activity 1, (**e**) Activity 2, and (**f**) Activity 3. An alpha value of 0.3 was applied to the plots (blue dots) to reduce the effect of over-plotting.

**Figure 4 sensors-24-05713-f004:**
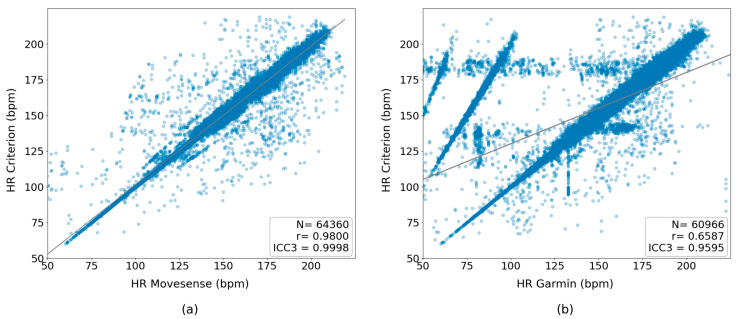
Correlation of the heart rate derived from the RR’ values through the different intensities of the incremental test: (**a**) Movesense HR+ vs. criterion and (**b**) Garmin HRM-Pro vs. criterion. An alpha value of 0.3 was applied to the plots (blue dots) to reduce the effect of over-plotting.

**Figure 5 sensors-24-05713-f005:**
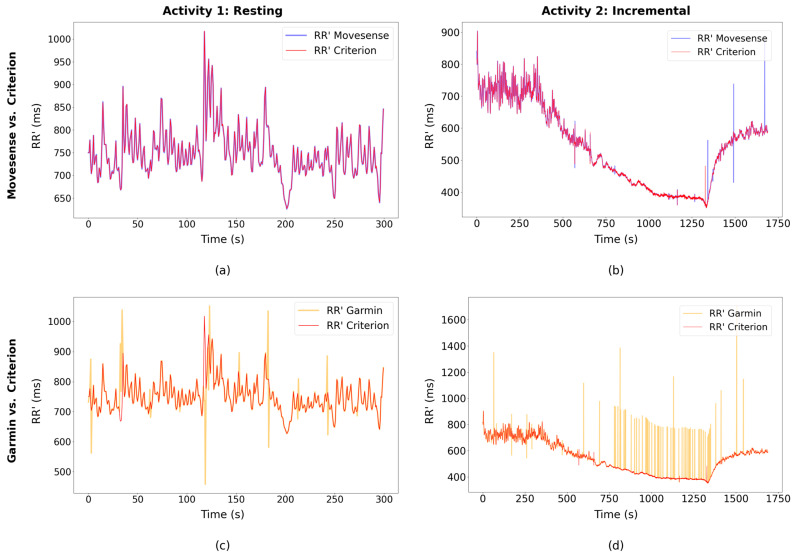
Comparative analysis of RR’ interval samples for one participant. (**a**) Activity 1 (resting), Movesense HR+ vs. 3-lead; (**b**) Activity 2 (incremental test), Movesense HR+ vs. criterion; (**c**) Activity 1 (resting), Garmin HRM-Pro vs. criterion; (**d**) Activity 2, Garmin HRM-Pro vs. criterion. Data for the same participant, activity and time are shown in (**a**) and (**c**) and in (**b**) and (**d**), respectively.

**Figure 6 sensors-24-05713-f006:**
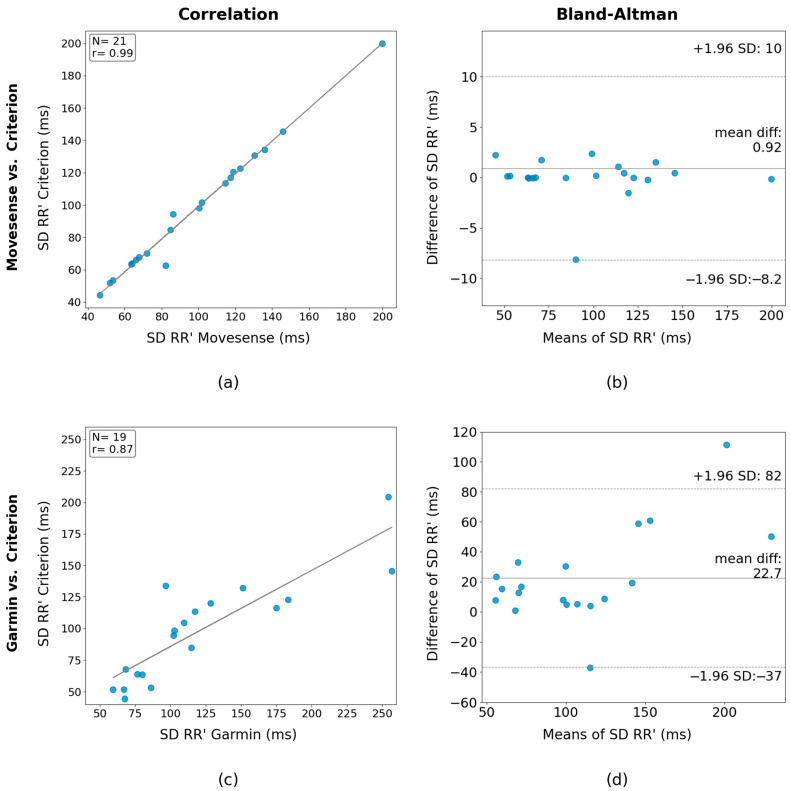
(**a**) Correlation of standard deviation (SD) of the RR’ intervals for Movesense HR+ and 3-lead ECG; (**b**) Bland-Altman plot of the SD of the RR’ intervals for Movesense HR+ and 3-lead ECG; (**c**) Correlation of SD of the RR’ intervals for Garmin HRM-Pro and 3-lead ECG; (**d**) Bland-Altman plot of the SD of the RR’ intervals for Garmin HRM-Pro and 3-lead ECG.

**Table 1 sensors-24-05713-t001:** Detail of experimental protocol.

Activity	Phase	Duration	Description	Speed/Power
Activity 1	Rest	5’	Participants were seated on a chair without engaging in conversation.	-
Activity 2	Warm-up	6’	A warm-up phase was initiated to prepare for the subsequent incremental ramp protocol.	8 km/h or 100 W
	Incremental	Until exhaustion *	Began at 8 km/h or 100 W, with increments of 1 km/h or 40 W (30 W for women) every 2 min until exhaustion.	increasing
	Cool-down	6’	After reaching exhaustion, participants were guided through a cool-down, returning to the warm-up speed/power.	8 km/h or 100 W
Activity 3	Final Rest	5’	Participants were again seated on a chair without talking, allowing a period of recovery.	-

* Exhaustion was determined when participants could not maintain the prescribed intensity.

**Table 2 sensors-24-05713-t002:** Equipment and materials.

Equipment Type	Model/Details	Metrics/Functionality	Sampling Rate
Treadmill	STEX 8020T	Programmed at different speeds	-
Cycloergometer	Lode Excalibur Sport	Programmed at different powers	-
3-lead ECG system (i.e., criterion)	3 Leads Shielded Bio Amp Cable, PowerLab 26T, ECG electrodes	Continuous ECG recording, interfaced with Labchart 8.1.24	1000 Hz
Flow direction mask	Turbine and an Arduino microcontroller	Continuous BF recording, interfaced with Labchart 8.1.24	1000 Hz
Movesense HR+ (single channel)	Model: OP174, with chest belt	ECG recording	500 Hz
Garmin HRM-Pro chest	Connected to Garmin Fenix 3 watch	HR, HRV, running dynamics	N/A

**Table 3 sensors-24-05713-t003:** Physiological parameters from the different activities.

Activity	Exercise	Number of Participants	Average RR’ (ms)	Average HR (bpm)	Average Max HR (bpm)	Average RR’ SD (ms)
Activity 1 (5 min rest)	Running	14	927.6	64.7	85.8	97.1
	Cycling	7	808.2	74.2	98.2	92.5
Activity 2 (Incremental exercise)	Running	11	390.3	153.7	201.5	53.0
	Cycling	7	416.5	144.1	192.0	64.8
Activity 3 (5 min rest)	Running	12	562.9	106.6	122.7	23.5
	Cycling	7	593.8	101.0	115.7	28.5

**Table 4 sensors-24-05713-t004:** Sensitivity and precision Movesense HR+.

Activity	Exercise	Participants	*TP*	*FN*	Sensitivity	*FP*	Precision
Activity 1 (Rest)	Running	14	4658	23	99.51%	27	99.42%
	Cycling	7	2669	3	99.89%	5	99.81%
Activity 2 (Exercise)	Running	11	40,163	175	99.56%	249	99.38%
	Cycling	7	23,988	33	99.86%	21	99.91%
Activity 3 (Rest)	Running	12	6463	31	99.52%	30	99.54%
	Cycling	7	3635	17	99.53%	16	99.56%

**Table 5 sensors-24-05713-t005:** Sensitivity and precision of Garmin HRM-Pro.

Activity	Exercise	Participants	*TP*	*FN*	Sensitivity	*FP*	Precision
Activity 1 (Rest)	Running	12	3658	176	95.41%	211	94.55%
	Cycling	7	2462	214	92.00%	288	89.53%
Activity 2 (Exercise)	Running	11	31,106	6059	83.70%	4005	88.59%
	Cycling	7	18,877	4924	79.31%	4382	81.16%
Activity 3 (Rest)	Running	12	5592	145	97.47%	57	98.99%
	Cycling	7	3261	116	96.56%	87	97.40%

## Data Availability

Data are available upon request to the corresponding author following an embargo period during PhD candidature. A free version of the software tools developed for this research is available through the links in the references.

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
