# Peer review of "Validity and Reliability of Movesense HR+ ECG Measurements for High-Intensity Running and Cycling"

_sensors, 2024, doi:10.3390/s24175713_

Round 1

Reviewer 1 Report (New Reviewer)

Comments and Suggestions for Authors

The manuscript describes the validity of Movesense HR+ electrocardiogram (ECG) data relative to traditional ECG systems. 

The present study therefore examined the reliability and validity of the Movesense HR+ ECG measurements in corporating an R-peak detection algorithm and an ECG data collection routine. The aim is to compare these Movesense HR+ ECG measure ments, across different exercise modes and intensities, with criterion measurements from a standard 3-lead ECG system. 

Some comments as follows.

In lines 149-150

Peak detection was verified through visual inspection of the ECG time-series plots and identified peaks, ensuring the accuracy of R-peak detection.

Q1. Peak detection should be verified through some algorithm not by visual inspection.

 In line 192-193

All data, including the corresponding R-peaks and the derived RR' intervals, were visually inspected to ensure accuracy and remove artifacts.

Q2. How can you  visually inspected to ensure accuracy and remove artifacts?

In line 144-147

The ECG data from the 3-lead and Movesense HR+ device were consequently processed [5]using the self-created peak detection algorithm, Figure 2. The algorithm utilises various techniques such as adaptive thresholding, R and S peak detection, waveform analysis, abnormal peak detection, and iteration to optimise ECG peak detection during exercise.

Q3.  The algorithm should be described more detaily.

Comments on the Quality of English Language

Quality of English Language needs to be improved.

Author Response

  1. In lines 149-150

Peak detection was verified through visual inspection of the ECG time-series plots and identified peaks, ensuring the accuracy of R-peak detection.

Q1. Peak detection should be verified through some algorithm not by visual inspection.

We agree that algorithmic methods should be used for peak detection. Multiple algorithms were used independently then convergently to detect and verify peaks, as detailed in our response to Question 3 below. The accuracy and reliability of those algorithms was further checked by visual inspection using a customised graphical user interface (GUI). The GUI displays ECG voltages and waveforms alongside the detected R-peaks for each participant, activity, and device. Additionally, the GUI presents markers for all abnormal beats, allowing us to verify potential artifacts or incorrect detections. A simplified executable version of the GUI, along with testing examples, is available via a link in the references [9]. A video tutorial is also provided, demonstrating how to use the GUI to open Movesense, Criterium, or Garmin files, synchronise the global timestamp, and plot the voltages with the R-peaks. This visual validation method complements our algorithmic detection, helping ensure high accuracy in R-peak identification.

[9]        R. Martin Gomez, “Article Resources (GUI and samples).” Accessed: May 26, 2024. [Online]. Available: https://drive.google.com/drive/folders/1kkTDTgPSZE54BcgcYSdfB6KpY7cFPZsf?usp=sharing

  1. In line 192-193

All data, including the corresponding R-peaks and the derived RR' intervals, were visually inspected to ensure accuracy and remove artifacts.

Q2. How can you visually inspected to ensure accuracy and remove artifacts?

As mentioned in our response to Question 1, we utilised a customised GUI for visual inspection. Visual validation, supported by markers for abnormal beats, is a widely accepted method for testing ECG samples and validating R-peak detection algorithms.

Although our algorithm has been successfully tested with other sample ECG tests, this information is beyond the main scope of our article. In this study, accuracy was ensured using thorough visual inspection of all results, which we believe is sufficient to validate the reliability of the Movesense HR+ ECG measurements compared to the Garmin HRM.

  1. In line 144-147

The ECG data from the 3-lead and Movesense HR+ device were consequently processed [5]using the self-created peak detection algorithm, Figure 2. The algorithm utilises various techniques such as adaptive thresholding, R and S peak detection, waveform analysis, abnormal peak detection, and iteration to optimise ECG peak detection during exercise.

Q3. The algorithm should be described more detaily.

To clarify the R peak detection algorithm in the manuscript, the paragraph has been updated with more detail:

Revised Manuscript Text, on lines 151-168: " The algorithm for R-peak detection involves several steps to ensure accurate and reliable peak identification during exercise. The first step is to divide the ECG data into intervals, calculating the data frequency and managing the intervals based on this frequency. Next, two independent peak detection processes are performed for each interval: one for R peaks and one for S peaks, with the ECG data inverted for the latter. Each detection method involves sorting the data by voltage and iteratively determining a threshold for peak detection by analysing the highest voltage values. After identifying the R and S peaks, the results from both methods are combined to form a comprehensive list, filtering and merging peaks detected by both methods. Subsequently, the algorithm performs a statistical analysis of the detected peaks identifying abnormal RR intervals. For these intervals, new potential peaks are detected by analysing subsets of ECG data around target timestamps and calculating associated characteristics. The algorithm iteratively optimises its parameters, recalculates heart rate and heart rate variability, and refines the detection process to maintain high accuracy. Finally, the detected peaks are compiled, and additional statistical analyses are conducted to ensure data quality. This comprehensive approach allows the algorithm to handle the complexities of ECG signal processing during physical activity effectively, focusing on providing robust and accurate results for further analysis."

Reviewer 2 Report (New Reviewer)

Comments and Suggestions for Authors

Can the authors elaborate how the results could be affected by the different position of the Movesense HR+ and Garmin HRM?

There are 2 Figure reference errors that should be corrected.

Author Response

Comment 1: Can the authors elaborate how the results could be affected by the different position of the Movesense HR+ and Garmin HRM?

Response 1: Although we conducted preliminary tests with both the Movesense HR+ and Garmin HRM-Pro in different positions and observed no significant differences in the results, we chose not to include this information in the results section. This decision was made because we believe a dedicated study would be required to thoroughly investigate the impact of device positioning on measurement accuracy. We acknowledge this limitation (Lines 418-423), and have refined the wording of the purpose statement to reflect the primary and secondary aim; i.e., abstract (lines 10-11) and Introduction (lines 67-68).

The primary objective of our study was the validation of the Movesense HR+, and thus, we consistently placed the Movesense HR+ under the nipples for all participants, as detailed in the methods section on lines 90-94. The Garmin HRM-Pro's placement varied based on participant comfort and feasibility, especially for female participants wearing sports bras. The skin was cleaned with alcohol before attaching the electrodes, and no glue was used; instead, the electrodes were wetted with water to simulate typical device usage.

By clearly stating the placement of the chest straps, we aimed to maintain methodological clarity and focus on the primary goal of validating the Movesense HR+. Future studies could explore the effects of different positioning in greater detail to provide a comprehensive understanding of any potential variations in measurement outcomes.

Comment 2: There are 2 Figure reference errors that should be corrected.

Response 2: We added references to Figure 2 on lines 255 and 266 to indicate the figure containing the formulas used for the results presented in section 3.1. To avoid confusion, we have removed these references and retained only the reference to the table presenting the most relevant results of the analysis.

Reviewer 3 Report (New Reviewer)

Comments and Suggestions for Authors

Dear authors,

I present my suggestions and evaluation of your manuscript “Validation and Reliability of Movesense HR+ ECG Measurements: A Comparative Analysis with Garmin HRM” below.

The overall merit of the proposed manuscript is good. I am recommending reconsideration after major revision. I am adding the main points and questions that need to be solved or clarified.

Questions and Quotes

  1. There are presented several values in abstract - mean differences (SD) in RR’ intervals of 0.23 16 (44) ms for Movesense HR+ at rest and 0.38 (37) ms during the incremental test. Mean difference for 17 Garmin HRM at rest was -8.5 (210) ms and for the incremental test 27.7 (280) ms. The values in brackets should represent SD but they are high in comparison to the mean values. Are they in the same units? How can you explain this?

  2. The dataset is small and imbalanced in the case of men-to-woman comparison and in the case of cycling to running as well. 

  3. The authors present the parameters of ROC analysis like negative predictive value, detection error rate, and so on, but then only the PPV and sensitivity are presented. Why? 

  4. How the correlation was computed?

  5. Line 333 - I don't understand how this can improve ecological validity. Can you explain this more in detail?

  6. There are several grammatical and typographical mistakes and errors in the manuscript (for example line 310.

  7. What was the main goal and what was the main output? These points need to be highlighted in the manuscript.

  8. The ORCID should be added to all of the authors. 

Author Response

  1. There are presented several values in abstract - mean differences (SD) in RR’ intervals of 0.23 16 (44) ms for Movesense HR+ at rest and 0.38 (37) ms during the incremental test. Mean difference for 17 Garmin HRM at rest was -8.5 (210) ms and for the incremental test 27.7 (280) ms. The values in brackets should represent SD but they are high in comparison to the mean values. Are they in the same units? How can you explain this?

We wanted to keep the values in the abstract consistent with the results presented in Figure 3 on lines 271-272, which is why the values in brackets represented the upper confidence interval, 1.96 * standard deviation (SD), used in Bland-Altman analysis. However, the SD values of that sentence in the abstract has been corrected to reflect the SD instead of the upper confidence interval. New text on lines 16-19:

“Bland-Altman analysis compared to the criterion indicated mean differences (SD) in RR’ intervals of 0.23 (22.3) ms for Movesense HR+ at rest and 0.38 (18.7) ms during the incremental test. Mean difference for Garmin HRM at rest was -8.5 (111.5) ms and for the incremental test 27.7 (128.7).”

The high values of the SD compared with the mean demonstrate how these devices can provide very accurate average values of the RR’ intervals or Heart Rate, while having difficulties detecting every single value accurately, especially in the case of Garmin. Sample size calculations are not agreed upon for validity and reliability, as they are generally for null hypothesis significance testing.

  1. The dataset is small and imbalanced in the case of men-to-woman comparison and in the case of cycling to running as well.

We believe that the sample size, sex distribution, and mode distribution are all suitable for the purposes of this experiment. The purpose was not to compare the effects of sex or mode of exercise, but rather to obtain data from a range of healthy participants, across the full range of cardiorespiratory response, in each of these two common modes of dynamic exercise. Sample size calculations are not agreed upon in the literature for studies such as ours in which time-continuous measures (ECG voltage and RR’ intervals) are assessed for reliability and validity.

  1. The authors present the parameters of ROC analysis like negative predictive value, detection error rate, and so on, but then only the PPV and sensitivity are presented. Why?

To simplify the tables, we included the most relevant results from the study: Sensitivity and Precision. The detection error rates, specificity, and accuracy were directly presented in the text. In the Results section on lines 244-246, the manuscript states: “With a total TN of 38,059,993 for the three activities, the resulting overall DER, Specificity, and Accuracy of Movesense HR+ were 0.002%, 99.999%, and 99.998%, respectively.” On lines 255-256, the manuscript states: “With a total TN of 37,454,530 for the three activities, the resulting overall DER, Specificity, and Accuracy of Garmin HRM were 0.055%, 99.969%, and 99.945%, respectively.”

  1. How the correlation was computed?

The correlation between the RR' intervals from the Movesense HR+ and the Criterium 3 lead ECG was computed using the Pearson correlation coefficient. We used the pearsonr function from the SciPy library in Python. To compute the correlation, we passed two lists of RR' intervals to the pearsonr function: one list containing the intervals from the device (Movesense HR+ or Garmin HRM) and the other from the Criterium 3-lead ECG. The function then returns the correlation coefficient r. Both sets of RR' intervals had the same start and end global timestamps.

  1. Line 333 - I don't understand how this can improve ecological validity. Can you explain this more in detail?

The term "ecological validity" refers to the extent to which the findings of a study can be generalized to real-world settings. In this context, the use of portable Bluetooth Low Energy (BLE) devices, such as the Movesense HR+ and Garmin HRM-Pro, allows for the collection of heart rate and ECG data outside of controlled laboratory environments.

By using these portable devices, researchers can gather data in more natural and varied settings, reflecting the participants' typical activities and conditions. This transition from lab to real-world settings enhances the ecological validity of the metrics obtained, as the data are more representative of actual daily life conditions, thus providing more robust and applicable insights.

To clarify the text in the manuscript, we have revised it as follows (Lines 353-355):

Revised Manuscript Text: "Moreover, the use of portable BLE devices can help transition research from controlled lab environments to real-world settings, improving the reliability and applicability of metrics obtained."

This revised text maintains the original meaning while hopefully now providing a clearer explanation of how portable BLE devices enhance the applicability of the study's findings to real-world scenarios.

  1. There are several grammatical and typographical mistakes and errors in the manuscript (for example line 310.

Previous text:

“Figure 6 shows correlation and Bland-Altman plots of Movesense HR+ and Garmin HRM against the 3-lead for all the participants.”

Correction on lines 330-332:

“Figure 6 shows correlation and Bland-Altman plots of the Movesense HR+ and Garmin HRM against the 3-lead for all participants.”

We have subsequently proof-read the entire manuscript and made minor grammatical changes.

  1. What was the main goal and what was the main output? These points need to be highlighted in the manuscript.

The main goal of our study was to establish the validity and reliability of the Movesense HR+ ECG measurements compared to the criterion 3-lead ECG and the industry leader Garmin HRM. As stated in the Abstract on lines 10-12: "The purpose of this study was to establish the validity and reliability of Movesense HR+ ECG measurements compared to the criterion 3-lead ECG and industry leader Garmin HRM."

We have updated the previous paragraph in the Introduction on lines 60–62 “The aim is to compare these Movesense HR+ ECG measurements, across different exercise modes and intensities, with criterion measurements from a standard 3-lead ECG system.” to the following, lines 53–55: “The primary aim of this study was therefore to compare the validity and reliability of Movesense HR+ ECG measurements across different exercise modes and intensities, using criterion measurements from a standard 3-lead ECG system.” And on lines 65–67 “A secondary aim of this study was to do a reference comparison for the industry-leading device, Garmin HRM-Pro, against the ECG criterion measurements.”

In the Conclusion section on lines 427-437 the manuscript states: “Our evaluation of Movesense HR+ ECG data, coupled with the flexibility and potential for refinement offered by ECG voltage-based systems highlights its potential as a portable reliable tool for accurate ECG data recording across various exercise modalities. These findings have significant implications for both research studies and individual health monitoring, and development of enhanced physiological assessments of human performance. Users can make use of the knowledge gained from this study to further develop their use of Movesense HR+ and other ECG-based wearable devices. This includes ensuring optimal sensor placement, utilising real-time monitoring features to verify data accuracy during collection, and employing post-processing software for accurate calculation and validation of metrics derived from the raw ECG data, especially during intense exercise."

In the Discussion section on lines 392-397 the manuscript states: “The implementation of the FreeLab software environment, customised firmware, Android apps, and Python GUI has demonstrated its potential to support peak calculations and further analysis. The ability to synchronize ECG data collected from Movesense or a 3-lead system with HRV/RR' data from .fit files (e.g., Garmin's .fit files) enhances its utility for research analysis and comparison studies [2], [3], [5], [6].”

  1. The ORCID should be added to all of the authors. 

ORCID identification has been added for all of the authors.

This manuscript is a resubmission of an earlier submission. The following is a list of the peer review reports and author responses from that submission.

Round 1

Reviewer 1 Report

Comments and Suggestions for Authors

Dear Authors,

Thank you for submitting this article. It has a nice flow to the readers. I would appreciate it if you can please address the following comments:

1. In light of your study's findings, what particular scenarios or real-world uses do you see for the Movesense HR+ ECG device?

2. Could you provide more details about the possible ramifications of your research findings for individual health monitoring and research studies? How do these results add to the body of knowledge already known in the field?

3. How did you make sure the Movesense HR+ ECG data collection method was accurate and dependable in a range of exercise intensities and modes?

4. Could you elaborate on the reasoning for your study's selection of young, healthy participants? If there were people in the sample who had other health conditions or demographic traits, how could the results change?

5. You said that when compared to the Garmin HRM, the Movesense HR+ showed great agreement with 3-lead ECG. Could you elaborate on the reasons behind the observed performance difference?

6. In comparing the Movesense HR+ with other devices, what unique difficulties did you run across, especially during periods of intensive exercise? In your analysis, how did you resolve or lessen these issues?

How likely is it that the Movesense HR+ device will be widely adopted in different contexts, taking into account aspects like affordability, accessibility, and ease of use?

8. In light of the results of your study, what restrictions or limits should practitioners and researchers be aware of while using the Movesense HR+ device? What potential effects can these drawbacks have on their applicability in actual situations?

9. In future research, which particular areas or facets of the Movesense HR+ device's functionality would you suggest receiving more attention or being improved upon?

10. How might future technological or methodological developments improve the functionality and precision of ECG-based wearables like Movesense HR+?

Reviewer 2 Report

Comments and Suggestions for Authors

Summary

This paper used three different brands of commercially available ECG/Heart Rate Monitor devices to collect data from 21 healthy adults during rest and specific activities (running or cycling). The utilized devices include ADInstruments standard 3-lead ECG, Movesense HR+, and Garmin HR. The performance of the Movesense HR+ was evaluated compared to the performance of the other two systems.

General Comments

Overall, the manuscript appears well presented and reads fluently; however, it lacks technical/scientific contents to be published in a high-quality technical journal.   

The paper lacks reference to scientific work by other research groups to motivate the audience and justify the importance of the work, explain standard methods/procedures, compare the results, and draw conclusions.

The paper heavily relies on text to explain the methods/procedures. The journal guidelines encourage authors to utilize equations, diagram, or charts to communicate information. For instance, a simple concept such as absolute or relative timestamps in Linux can be explained by a simple equation rather than lengthy descriptions in several paragraphs.

Use an appropriate unit of time (such as ms) for RR’ results in the paragraphs, tables, and figures.

The results section failed to explain why the RR’ results for the Movesense HR+ and Garmin devices are inferred from the data by the standard 3-lead ECG system. If Movesense HR+ and Garmin HR are sold as standalone heart rate monitor devices, they should be capable of providing such information.

The results section needs to present the voltage vs. time plots for the standard 3-lead ECG, Movesense HR+, and Garmin devices during rest and activities.

Specific Comments

Lines 27-55:
Appropriate citations are needed to demonstrate the importance of the field, relevance of the proposed research, and authors’ adequate research into the field. In a technical journal, please avoid description of products and their features using a marketing tone/style.  

Lines 82-100:
The equipment and features can be presented in a tabular form.
The physical features of the sensors/devices need to be clearly explained. For instance,

·        Clarify the difference between a 3-channel device and a 3-lead device!

·        What was the sampling rate of the Garmin device?

·        How many electrodes do the Movesence HR+ and Garmin devices rely on?

·        What type of electrodes does each device use, and what are the advantages or disadvantages?

Lines 100-151:
Section 2.4 is unnecessarily filled with the details about the stored data file formats. Such irrelevant information may be interpreted as space fillers.

Lines 111-115:
Referral to the IMU data could be misleading as the IMU data was not in the scope of this paper and its analysis. Please omit it to avoid ambiguities.

Line 117:
Explain how a 2-lead device can generate “16 voltage values corresponding to different ECG leads”.  

Line 129:
Express the “specific mathematical transformation”.

Lines 131-132:
Please clarify what the “self-created peak detection algorithm” is!

Line 150:
Specify the sampling rate for the Garmin device.

Line 162:
The term “Num_R_ peaks_device_3lead” is misleading; revise it for clarity.

Line 170:
Explain the rationale for the chosen 100 ms threshold.

Lines 237-238:
The authors need to explain what a standard Bland-Altman plot is and clarify their deviations from the standard method presenting the RR’ 3-lead data on the horizontal axis and the RR’ of Movesense/Garmin device on the vertical axis.

Comments on the Quality of English Language

Lines 191-192:
Revise the sentence for clarity.